# Hold me tight! Influence of discriminative features on deep network boundaries

**Guillermo Ortiz-Jiménez**[*]
EPFL, Lausanne, Switzerland
guillermo.ortizjimenez@epfl.ch

**Apostolos Modas**[*]
EPFL, Lausanne, Switzerland
apostolos.modas@epfl.ch

**Seyed-Mohsen Moosavi-Dezfooli**
ETH Zürich, Zurich, Switzerland
seyed.moosavi@inf.ethz.ch

**Pascal Frossard**
EPFL, Lausanne, Switzerland
pascal.frossard@epfl.ch

## Abstract

Important insights towards the explainability of neural networks reside in the characteristics of their decision boundaries. In this work, we borrow tools from the field of adversarial robustness, and propose a new perspective that relates dataset features to the distance of samples to the decision boundary. This enables us to carefully tweak the position of the training samples and measure the induced changes on the boundaries of CNNs trained on large-scale vision datasets. We use this framework to reveal some intriguing properties of CNNs. Specifically, we rigorously confirm that neural networks exhibit a high invariance to non-discriminative features, and show that the decision boundaries of a DNN can only exist as long as the classifier is trained with some features that hold them together. Finally, we show that the construction of the decision boundary is extremely sensitive to small perturbations of the training samples, and that changes in certain directions can lead to sudden invariances in the orthogonal ones. This is precisely the mechanism that adversarial training uses to achieve robustness.

## 1 Introduction

The set of points that partitions the input space onto labeled regions is known as the *decision boundary* of a classifier. Describing how a classifier creates such boundaries is crucial for its explainability. Interestingly, even when deep networks succeed on a task, their high vulnerability to imperceptible perturbations [1, 2] implies that their boundaries lie alarmingly close to any input sample. This unintuitive behaviour contradicts the common belief that a successful classifier should be invariant to non-discriminative information of its input data. However, it seems that such perturbations are not irrelevant signals, but rather discriminative features of the training set [3, 4].

In that sense, explaining the mechanisms that construct the decision boundary of deep neural networks is key to understand the dynamics of adversarial training [5]. This training scheme only differs from standard training in that it slightly perturbs the training samples during optimization. However, these small changes can utterly change the geometry of these classifiers [6].

An example of such change can be seen in Fig. 1, which shows the minimal perturbations – constrained to lie on a low and a high frequency subspace – required to flip the decision of a network. The norm of the perturbations measures the distance (margin) to the decision boundary in these subspaces. Clearly, reaching the boundary using high frequency perturbations requires much more energy than using

---

[*]Equal contribution. Correspondence to {guillermo.ortizjimenez, apostolos.modas}@epfl.ch.
The code to reproduce our experiments can be found at https://github.com/LTS4/hold-me-tight.

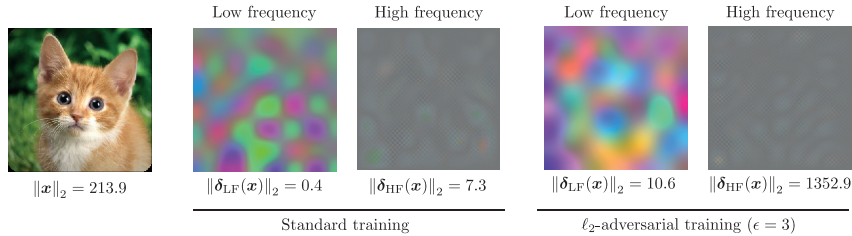

Figure 1: Minimal adversarial perturbations constrained to lie in different DCT frequency bands ($8 \times 8$ subspaces taken from the top left and bottom right of the $224 \times 224$ DCT matrix) for a ResNet-50 trained (**left**), and adversarially trained (**right**) on ImageNet.

low frequency ones [7]. But surprisingly, when the network is adversarially trained [5], the largest increase in margin happens in the high frequency subspace. Note that, on the standard network, this distance is already much greater than the size of the training perturbations. Based on this observation, we pose the following questions:

1. *How is the margin in different directions related to the features in the training data?*
2. *How can very small perturbations significantly change the geometry of deep networks?*

In this work, we propose a novel approach to answer these questions. In particular, we develop a new methodology to construct a local summary of the decision boundary of a neural network from margin observations along a sequence of orthogonal directions. This framework permits to carefully tweak the properties of the training samples and measure the induced changes on the boundaries of convolutional neural networks (CNNs) trained on synthetic and large-scale vision datasets (e.g., ImageNet). The main contributions of our work are the following:

1. We provide a new perspective on the relationship between the distance of a set of samples to the boundary, and the discriminative features used by a network. We empirically support our findings by extensive evaluations on both synthetic and real datasets.

2. Via a series of carefully designed experiments, we rigorously confirm the "common belief" that CNNs tend to behave as ideal classifiers and are approximately invariant to non-discriminative features of a dataset.

3. We further show that the construction of the decision boundary is extremely sensitive to the position of the training samples, such that very small perturbations in certain directions can utterly change the decision boundaries in some orthogonal directions.

4. Finally, we demonstrate that adversarial training exploits this training sensitivity and invariance bias to build robust classifiers.

We believe that the perspective proposed in this paper can have implications in future research on explainability and robustness, as it gives a new way to measure and understand decision boundaries. This new framework can be used to shed light onto the dynamics and inductive bias of deep learning.

**Related work** Since the publication of [8], a big body of research has focused on understanding the inductive bias of deep networks as a way to explain generalization in deep learning [9, 10]. Remarkably, for linear classifiers, optimizing a logistic loss using gradient descent is equivalent to maximizing margin in the input space [11]. Furthermore, for deep networks, some recent results suggest that margin is maximized in the logit space [12].

Interestingly, recent works have established the link between adversarial perturbations and discriminative features of the training sets [3, 4, 13]. This has led to the conjecture that, in most datasets, there exist robust and non-robust features that neural networks exploit to construct their decision boundaries. What exactly are these features, and how do networks construct these boundaries is however not addressed by these authors. In this sense, the authors of [14] argue that the excessive invariance in the boundaries introduced by adversarial training can explain its induced decrease in accuracy. In this work, we shed light on these phenomena by describing the strong inductive bias of the networks towards invariance to non-discriminative features, and the sensitivity of training to small perturbations.

The geometric properties of the decision boundaries of deep networks have previously been studied, mainly focused on their curvature [6, 15], and their decision region topology [15, 16]. From a robustness perspective, the distance to the boundary has been exploited to detect adversarial examples [17] and predict generalization gap [18, 19]. Furthermore, the unusual robustness of deep networks in certain frequencies [7, 20, 21] has recently been described. In this work, however, we give a constructive explanation for this phenomenon based on the role of dataset features in shaping the margins along different frequencies.

## 2  Proposed framework

Let $f : \mathbb{R}^D \to \mathbb{R}^L$ be the final layer of a neural network (i.e., logits), such that, for any input $x \in \mathbb{R}^D$, $F(x) = \operatorname{argmax}_k f_k(x)$ represents the decision function of that network, where $f_k(x)$ denotes the $k$th component of $f(x)$ that corresponds to the $k$th class. The decision boundary between classes $k$ and $\ell$ of a neural network is the set $\mathcal{B}_{k,\ell}(f) = \{x \in \mathbb{R}^D : f_k(x) - f_\ell(x) = 0\}$ (in general, we will omit the dependency with $k, \ell$ for simplicity). Unless stated otherwise, we assume that all networks are trained using a cross-entropy loss function and some variant of (stochastic) gradient descent. We also assume that training has been conducted for many epochs, and that it has approximately converged to a local minimum of the loss, achieving 100% accuracy on the training data [8][1].

In this work, we study the role that the training set $\mathcal{T} = \{(x^{(i)}, y^{(i)})\}_{i=0}^{N-1}$ has on the boundary $\mathcal{B}(f)$. Specifically, we propose to use adversarial proxies to measure the distribution of distances to the decision boundary along a sequence of well defined subspaces. The main quantities of interest are:

**Definition 1** (Minimal adversarial perturbations). *Given a classifier $F$, a sample $x \in \mathbb{R}^D$, and a sub-region of the input space $\mathcal{S} \subseteq \mathbb{R}^D$, we define the ($\ell_2$) minimal adversarial perturbation of $x$ in $\mathcal{S}$ as $\delta_{\mathcal{S}}(x) = \operatorname{argmin}_{\delta \in \mathcal{S}} \|\delta\|_2 \quad s.t. \quad F(x + \delta) \neq F(x)$.*

*In general, we will use $\delta(x)$ to refer to $\delta_{\mathbb{R}^D}(x)$.*

**Definition 2** (Margin). *The magnitude $\|\delta_{\mathcal{S}}(x)\|_2$ is the margin of $x$ in $\mathcal{S}$.*

Our main objective is to obtain a local summary of $\mathcal{B}(f)$ around a set of observation samples $\mathcal{O} = \{x^{(i)}\}_{i=0}^{M-1}$ by measuring their margin in a sequence of distinct subspaces $\{\mathcal{S}_j\}_{j=0}^{R-1}$. In practice, we use a subspace-constrained version of DeepFool [22][2] to approximate the margins in each $\mathcal{S}_j$.

In the adversarial robustness literature, DeepFool is generally regarded as one of the most efficient methods to identify minimal adversarial perturbations. Because we want to measure margin, norm-constrained attacks like PGD [5] are not suitable for our study. Besides, more complex attacks like C&W [23], or, even, using unconstrained gradient descent in the input space, are computationally much more demanding and harder to tune than DeepFool. Since they in general find very similar adversarial perturbations as DeepFool, we decided to opt for DeepFool in our work.

## 3  Margin and discriminative features

As known from previous studies on the robustness of deep learning [24], the distance from a sample to the boundary of a neural network can greatly vary depending on the search direction. This behaviour is generally translated into classifiers with small margins along some directions, and large margins along the others. In this section, we show that the small margin directions are associated with discriminative directions, and we provide a constructive procedure to identify them. This helps us to shed new light into the inductive bias of the training dynamics of neural networks.

### 3.1  Evidence on synthetic data

We want to show that neural networks only construct boundaries along discriminative features, and that they are invariant in every other direction[3]. To this end, we generate a balanced training set

$\mathcal{T}_1(\epsilon, \sigma)$ by independently sampling $N$ points $\boldsymbol{x}^{(i)} = \boldsymbol{U}(\boldsymbol{x}_1^{(i)} \oplus \boldsymbol{x}_2^{(i)})$ such that $\boldsymbol{x}_1^{(i)} = \epsilon y^{(i)}$ and $\boldsymbol{x}_2^{(i)} \sim \mathcal{N}(0, \sigma^2 \boldsymbol{I}_{D-1})$, where $\oplus$ denotes the concatenation operator, $\epsilon > 0$ the feature size, and $D = 100$. The labels $y^{(i)}$ are uniformly sampled from $\{-1, +1\}$. The multiplication by a random orthonormal matrix $\boldsymbol{U} \in \mathrm{SO}(D)$ is performed to avoid any possible bias of the classifier towards the canonical basis. Note that this is a linearly separable dataset with a single discriminative feature parallel to $\boldsymbol{u}_1$ (i.e., first row of $\boldsymbol{U}$), and all other dimensions filled with non-discriminative noise.

To evaluate our hypothesis, we train a heavily overparameterized multilayer perceptron (MLP) with 10 hidden layers of 500 neurons using SGD (test: $100\%$). Table 1 shows the margin statistics on the linearly separable direction $\boldsymbol{u}_1$; its orthogonal complement $\mathrm{span}\{\boldsymbol{u}_1\}^\perp$; a fixed random subspace of dimension S, $\mathcal{S}_{\mathrm{rand}} \subset \mathbb{R}^D$; and a fixed random subspace of the same dimensionality, but orthogonal to $\boldsymbol{u}_1$, $\mathcal{S}_{\mathrm{orth}} \subset \mathrm{span}\{\boldsymbol{u}_1\}^\perp$. From these values we can see that along the direction where the discriminative feature lies, the margin is much smaller than in any other direction. Therefore, we can see that the classification function of this network is only creating a boundary in $\boldsymbol{u}_1$ with median margin $\epsilon/2$, and that it is approximately invariant in $\mathrm{span}\{\boldsymbol{u}_1\}^\perp$.

Comparing the margin values for $\mathcal{S}_{\mathrm{orth}}$ and $\mathcal{S}_{\mathrm{rand}}$ we see that, if the observation basis is not aligned with the features exploited by the network, the margin measurements might not be able to separate the small and large margin directions. Indeed, since $\mathcal{S}_{\mathrm{orth}}$ is orthogonal to the only discriminative direction $\boldsymbol{u}_1$ we see that the margin values reported in this region are much higher than those reported in $\mathcal{S}_{\mathrm{rand}}$. The reason for this is that the margin required to flip the label of a classifier in a randomly selected subspace is of the order of $\sqrt{S/D}$ with high probability [24], and hence the non-trivial correlation of a random subspace with the discriminative features will always hide the differences between small and large margin directions.

Table 1: Margin statistics of an MLP trained on $\mathcal{T}_1(\epsilon = 5, \sigma = 1)$ along different directions ($N = 10,000$, $M = 1,000$, $S = 3$).

|  | $\boldsymbol{u}_1$ | $\mathrm{span}\{\boldsymbol{u}_1\}^\perp$ | $\mathcal{S}_{\mathrm{ORTH}}$ | $\mathcal{S}_{\mathrm{RAND}}$ |
|---|---|---|---|---|
| 5-PERC. | 1.74 | 4.85 | 30.68 | 17.21 |
| MEDIAN | 2.50 | 12.36 | 102.0 | 27.90 |
| 95-PERC. | 3.22 | 31.60 | 229.5 | 80.61 |

Finally, the fluctuations in the values and the fact that the classifier is not completely invariant to $\mathrm{span}\{\boldsymbol{u}_1\}^\perp$ might indicate that the network has built a complex boundary. However, similar fluctuations and finite values in $\mathrm{span}\{\boldsymbol{u}_1\}^\perp$ would also be expected, even if the model was linear by construction and was perfectly separating the training data[4].

## 3.2 Evidence on real data

In contrast to the synthetic data, where the discriminative features are known by construction, the exact description of the features presented in *real* datasets is usually not known. In order to identify these features and understand their connection to the local construction of the decision boundaries, we apply the proposed framework on standard computer vision datasets, and investigate if deep networks trained on real data also present high invariance along the non-discriminative directions of the dataset.

In our study, we train multiple networks on MNIST [25] and CIFAR-10 [26], and for ImageNet [27] we use several of the pretrained networks provided by PyTorch [28][5]. Let $W, H, C$ denote the width, height, and number of channels of the images in those datasets, respectively. In our experiments we use the 2-dimensional discrete cosine transform (2D-DCT) [29] basis of size $H \times W$ to generate the observation subspaces. In particular, let $\boldsymbol{\mathcal{D}} \in \mathbb{R}^{H \times W \times H \times W}$ denote the 2D-DCT generating tensor, such that $\mathrm{vec}(\boldsymbol{\mathcal{D}}(i, j, :, :) \otimes \boldsymbol{I}_C)$ represents one basis element of the image space. We generate the subspaces by sampling $K \times K$ blocks from the diagonal of the DCT tensor using a sliding window with step-size $T$: $\mathcal{S}_j = \mathrm{span}\{\mathrm{vec}\,(\boldsymbol{\mathcal{D}}\,(j \cdot T + k, j \cdot T + k, :, :) \otimes \boldsymbol{I}_C)\ k = 0, \ldots, K - 1\}$.

The sliding window on the diagonal of the DCT gives a good trade-off between visualization abilities in simple one-dimensional plots, and a diverse sampling of the spatial spectrum of natural images, with a well-defined gradient flowing from low to high frequencies[6]. The DCT has a long application tradition in image processing due to its good approximation of the decorrelating transform (KLT) [30].

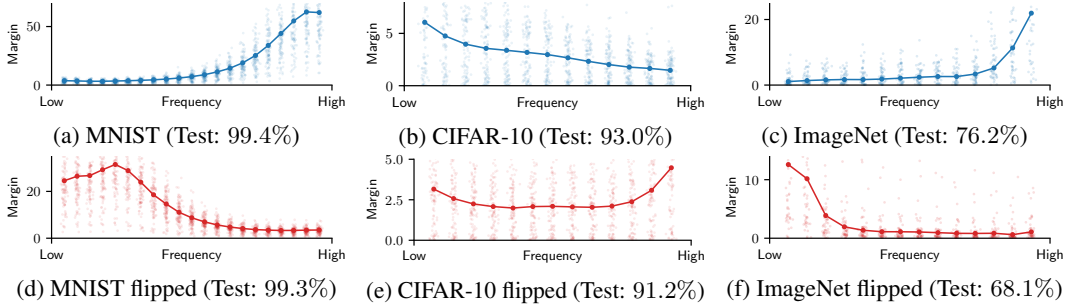

(a) MNIST (Test: 99.4%)     (b) CIFAR-10 (Test: 93.0%)     (c) ImageNet (Test: 76.2%)

(d) MNIST flipped (Test: 99.3%)    (e) CIFAR-10 flipped (Test: 91.2%)    (f) ImageNet flipped (Test: 68.1%)

Figure 2: Margin distribution of test samples in subspaces taken from the diagonal of the DCT (low to high frequencies). The thick line indicates the median values of the margin, and the shaded points represent its distribution. **Top**: (a) MNIST (LeNet) [25], (b) CIFAR-10 (DenseNet-121) [31] and (c) ImageNet (ResNet-50) [32] **Bottom**: (d) MNIST (LeNet), (e) CIFAR-10 (DenseNet-121) and (f) ImageNet (ResNet-50) trained on frequency "flipped" versions of the standard datasets.

Furthermore, in previous studies on the robustness of deep networks to different frequencies, the DCT was also the basis of choice [7] because it avoids dealing with complex subspaces.

We observe in practice that the DCT basis is also quite aligned to the features of these datasets, and hence it can give precise information about the discriminative features exploited by the networks. A more aligned basis with respect to the discriminative features would probably show a sharper transition between low and high margins. However, finding such network-agnostic bases is a challenging task without knowing the features *a priori*. The DCT is not perfectly feature-aligned, but it seems to be a good choice for comparing different architectures, especially if we compare its results to those obtained using a random orthonormal basis where differences in margin cannot be identified (c.f. Sec. N in Supp. material).

The margin distribution of the evaluated test samples is presented in the top of Fig. 2. For MNIST and ImageNet, the networks present a strong invariance along high frequency directions and small margin along low frequency ones. We will later show that this is related to the fact that these networks mainly exploit discriminative features in the low frequencies of these datasets. Notice, however, that for CIFAR-10 dataset the margin values are more uniformly distributed; an indication that the network exploits discriminative features across the full spectrum as opposed to the human vision system [33].

### 3.2.1 Adaptation to data representation

Towards verifying that the proposed framework can capture the relation between the data features and the local construction of the decision boundaries, we must first ensure that the direction of the observed invariance (large margin) is related to the features presented in the dataset, rather than being just an effect of the network itself.

Based on our observation that the margin tends to be small in low frequency directions and large in high frequency ones, we choose to carefully tweak the representation of the data, such that the low frequencies are swapped with the high frequencies. In practice, if $\mathfrak{D}$ denotes the forward DCT transform operator, the new image representation $\boldsymbol{x}'$ is expressed as $\boldsymbol{x}' = \mathfrak{D}^{-1}(\text{flip}(\mathfrak{D}(\boldsymbol{x})))$, where flip corresponds to one horizontal and one vertical flip of the DCT transformed image (see Fig. 3). Thus, if the direction of the resulting margin is strongly related to the data features, the constructed decision boundaries should also adapt to this new data

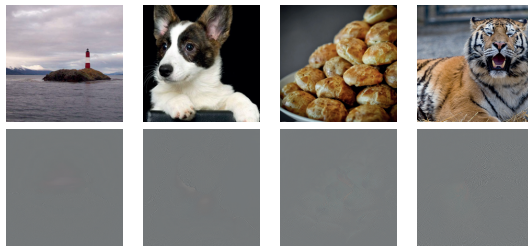

Figure 3: "Flipped" image examples from ImageNet. **Top**: original. **Bottom**: "flipped".

representation, and the margin along the invariant directions (high frequencies) should swap with the margin of the discriminative ones (low frequencies). Informally speaking, the margin distribution should "flip".

We apply our framework on multiple networks trained on the "flipped" datasets, and the margin distribution is depicted at the bottom of Fig. 2. For both MNIST and ImageNet, the directions of the decision boundaries indeed *follow* the new data representation – although they are not an exact mirroring of the original representation. This indicates that the margin strongly depends on the data distribution, and it is not solely an effect of the network architecture. Note again that for CIFAR-10 the effect is not as obvious, due to the quite uniform distribution of the margin.

### 3.2.2 Invariance and elasticity

The second property we need to verify is that the small margins reported in Fig. 2 do indeed correspond to directions containing discriminative features in the training set. For doing so, we use the insights of Fig. 2b on CIFAR-10 – where, opposed to the other datasets, we assume that there are exploited discriminative features in the whole spectrum – and show that, by explicitly modifying its features, we can induce a high margin response in the measured curve in a set of selected directions.

In particular, we create a low-pass filtered version of CIFAR-10 ($\mathcal{T}_{LP}$), where we retain only the frequency components in a $16 \times 16$ square at the top left of the diagonal of the DCT-transformed images. This way we ensure that no training image has any energy, hence information, outside of this frequency subspace. The median margin[7] of CIFAR-10 test samples for a network trained on $\mathcal{T}_{LP}$ is illustrated in Fig. 4. Indeed, by eliminating the high frequency content, we have forced the network to become invariant along these directions. This clearly demon-

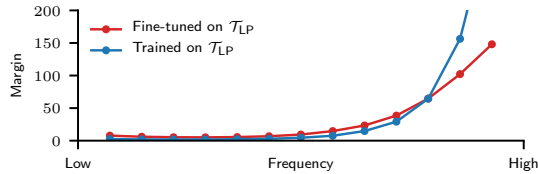

Figure 4: Median margin of test samples from CIFAR-10 for a DenseNet-121 (i) trained on CIFAR-10 and fine-tuned on $\mathcal{T}_{LP}$ (test: $90.79\%$), and (ii) trained on $\mathcal{T}_{LP}$ from scratch (test: $89.67\%$).

strates that there existed discriminative features in the high frequency spectrum of CIFAR-10, and that by effectively removing these from all the samples, the inductive bias of training pushes the network to become invariant to them.

Moreover, this effect can *also* be triggered during training. To show this, we start with the CIFAR-10 trained network studied in Fig. 2b and continue training it for a few more epochs with a small learning rate using only $\mathcal{T}_{LP}$. Fig. 4 shows the new median margins of this network. The fine-tuned network is again invariant to the high frequencies.

The elasticity to the modification of features during training gives a new perspective to the theory of catastrophic forgetting [34], as it confirms that the decision boundaries of a neural network can only exist for as long as the classifier is trained with the features that hold them together. In Sec. D of Supp. material, we provide an additional experiment to further discuss this relation in which we add and remove points from a dataset, thus triggering an elastic reaction in the network.

Finally, note that by training with only low frequency data, the test accuracy of the network on the original CIFAR-10 only drops around $3\%$[8]. Because $\mathcal{T}_{LP}$ has no high frequency energy, a network trained on it will uniformly extend its boundaries in this part of the spectrum and no high frequency perturbation will be able to flip the network's output. In contrast, testing $\mathcal{T}_{LP}$ data on a CIFAR-10 trained network only achieves $27.45\%$ test accuracy. This is because networks trained on CIFAR-10 do have boundaries in the high frequencies, and hence showing them original samples perturbed in this frequency range (i.e., $\mathcal{T}_{LP}$) can greatly change their decisions.

### 3.3 Discussion

The main claim in this section is that deep neural networks *only* create decision boundaries in regions where they identify discriminative features in the training data. As a result, there is a big

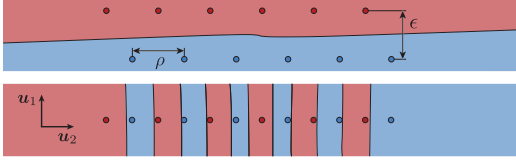 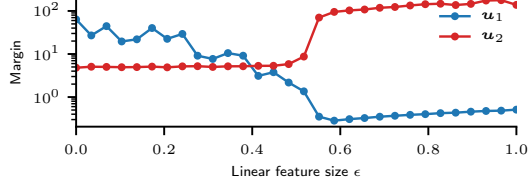

Figure 5: Cross-section of an MLP trained on $\mathcal{T}_2(\rho = 20, \epsilon, \sigma = 1, K = 3)$ with $\epsilon = 1$ (**top**) and $\epsilon = 0$ (**bottom**). Axes scaled differently.

Figure 6: Median margin values along $\boldsymbol{u}_1$ and $\boldsymbol{u}_2$ for MLPs (test: $100\%$ always) trained on $\mathcal{T}_2$ for different values of $\epsilon$ and $\rho = 20$.

relative difference in the large margin along the invariant directions, and the smaller margin in the discriminative directions.

The main difficulty for establishing causation in this idea is the fact that the discriminative features of real datasets are not known. Hence, determining their role on the geometry of a trained neural network can only be done by artificially manipulating the data. In particular, there are two main confounding factors that might alternatively explain our results: the network architecture or the training algorithm. However, the experiments in Sec. 3.2 are precisely designed to rule out their influence in this phenomenon.

Specifically, in the flipping experiments, flipping the data – *ceteris paribus* – also flips the margin distribution, thus demonstrating that the margins are necessarily caused by the information present in the data. The other interventions we do on the samples (e.g., low-pass experiments) confirm that, in the absence of information in a certain discriminative subspace, the network becomes invariant along this discriminative subspace. Therefore, we believe that there is indeed a causal connection between the features of the data and the measured margins in these neural networks. In fact, parallel theoretical studies have demonstrated that the ability of neural networks to distinguish between discriminative and non-discriminative noise subspaces in a dataset is one of the main advantages of deep learning over kernel methods [35].

## 4 Sensitivity to position of training samples

Our novel framework to relate boundary geometry and data features can help track the dynamics of learning. In this section, we use it to explain how training with a slightly perturbed version of the training samples can greatly alter the network geometry. We further analyze how adversarial training can be so successful in removing features with small margin to increase the network's robustness.

### 4.1 Evidence on synthetic data

We train multiple times an MLP with the same setup as in Section 3.1, but this time using slightly perturbed versions of the same synthetic dataset. In particular, we use a family of training sets $\mathcal{T}_2(\rho, \epsilon, \sigma, K)$ consisting in $N = 10,000$ independent $D = 100$-dimensional samples $\boldsymbol{x}^{(i)} = \boldsymbol{U}(\boldsymbol{x}_1^{(i)} \oplus \boldsymbol{x}_2^{(i)} \oplus \boldsymbol{x}_3^{(i)})$ such that $\boldsymbol{x}_1^{(i)} = \epsilon y^{(i)}$; $\boldsymbol{x}_2^{(i)} = \rho \cdot k$ when $y^{(i)} = +1$, and $\boldsymbol{x}_2^{(i)} = \rho \cdot (k + \frac{1}{2})$ when $y^{(i)} = -1$, where $k$ is sampled from a discrete uniform distribution with values $\{-K, \ldots, K-1\}$; and $\boldsymbol{x}_3^{(i)} \sim \mathcal{N}(0, \sigma^2 \boldsymbol{I}_{D-2})$ (see Fig. 5). Here, $\epsilon, \rho \geq 0$ denote the feature sizes. Again, the multiplication by a random orthonormal matrix $\boldsymbol{U} \in \mathrm{SO}(D)$ avoids any possible bias of the network towards the canonical basis. Note that for $\epsilon > 0$ this training set will always be linearly separable using $\boldsymbol{u}_1$, but without necessarily yielding a maximum margin classifier. Especially when $\rho \gg \epsilon$.

Fig. 6 shows the median margin values of $M = 1,000$ observation samples for an MLP trained on different versions of $\mathcal{T}_2(\rho, \epsilon, \sigma, K)$ with a fixed $\rho = 20$, but a varying small $\epsilon$. Based on this plot, it is clear that for very small $\epsilon$ the neural network predominantly uses the information contained in $\boldsymbol{u}_2$ to separate the different classes. Indeed, for $\epsilon < 0.2$, the network is almost invariant in $\boldsymbol{u}_1$, and it uses a non-linear alternating pattern in $\boldsymbol{u}_2$ to separate the data[9] (see bottom row of Fig. 5). On the contrary, at $\epsilon > 0.5$ we notice a sharp transition in which we see that the neural network suddenly changes its behaviour and starts to linearly separate the different points using only $\boldsymbol{u}_1$ (see top row of Fig. 5).

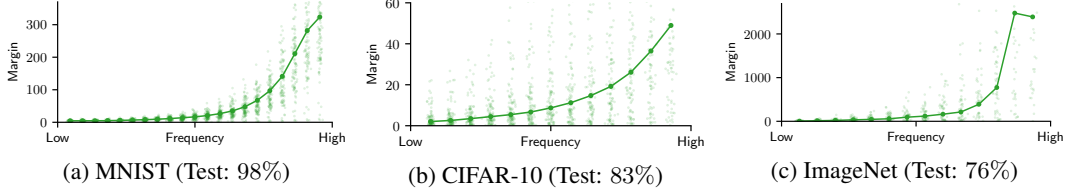

(a) MNIST (Test: 98%)     (b) CIFAR-10 (Test: 83%)     (c) ImageNet (Test: 76%)

Figure 7: Margin distribution of test samples in subspaces taken from the diagonal of the DCT (low to high frequencies). Adversarially trained networks using $\ell_2$ PGD [5] (a) LeNet (Adv: 76%), (b) DenseNet-121 (Adv: 55%) and (c) ResNet-50 (Adv: 35%).

We conjecture that this phenomenon is rooted on the strong inductive bias of the learning algorithm to build connected decision regions whenever geometrically and topologically possible, as empirically validated in [15]. Here, we go one step further and hypothesize that the inductive bias of the learning algorithm has a tendency to build classifiers in which every pair of training samples with the same label belongs to the same decision region. If possible, connected by a straight path.

We see Fig. 6 as a validation of this hypothesis. For small values of $\epsilon$, it is hard for the algorithm to find solutions that connect points from the same class with a straight path, as this is very aligned with $u_2$. However, there is a precise moment (i.e., $\epsilon = 0.5$) in which finding such a solution becomes much easier, and then the algorithm suddenly starts to converge to the linearly separating solution.

At this stage it is important to highlight that repeating the same experiment with a different random seed, or for a fixed initialization, does not affect the results. Furthermore, overfitting cannot be the cause of these results, as the MLP always achieves 100% test accuracy for $\epsilon < 0.5$, as well. Finally, adding a small weight decay (i.e., $10^{-3}$) does not help the network find the linearly separable solution for $\epsilon < 0.5$; it rather hinders its convergence (i.e., final train accuracy is 50%).

It remains unclear whether this inductive bias is the only mechanism that can trigger a sharp transition in the type of learned decision boundaries, or if there are other types of biases that can cause the same effect. In any case, we believe that the significant difference in the type of function that the algorithm learns when trained with very similar training samples (see Fig. 5), is an unambiguous confirmation of the sensitivity of deep learning to the exact position of its training input.

Concurrent work [36] has also used a similarly constructed dataset to $\mathcal{T}_2(\rho, \epsilon, \sigma, K)$ to argue that the simplicity bias of a neural network when trained using standard procedures might be responsible for the selection of non-robust features in the dataset [4].

## 4.2 Connections to adversarial training

Finally, we show that adversarial training exploits the type of phenomena described in Sec. 4.1 to reshape the boundaries of a neural network. In this regard, Fig. 7 shows the margin distribution across the DCT spectrum of a few adversarially trained networks[10]. As expected, the margins of the adversarially trained networks are significantly higher than those in Fig. 2.

Surprisingly, though, the largest increase can be noticed in the high frequencies for all datasets. Considering that adversarial training only differs from standard training in that it slightly moves the training samples, it is imperative that deep networks converge to very different solutions under such small modifications. The next experiments on CIFAR-10 shed light on the dynamics of this process.

**Adversarial perturbations can trigger invariance in orthogonal directions**   Slightly perturbing the training samples can remove features in an unpredictable manner. Fig. 8 shows the spectral decomposition of the adversarial perturbations crafted during adversarial training of CIFAR-10. The energy of the perturbations during training is always concentrated in the low frequencies, and has hardly any high frequency content. However, the greatest effect on margin is seen on the orthogonal high frequency directions (see Fig. 7). This is similar to what is seen in Fig. 5, where slightly perturbing the training samples along $u_1$ drastically affects the margin along $u_2$.

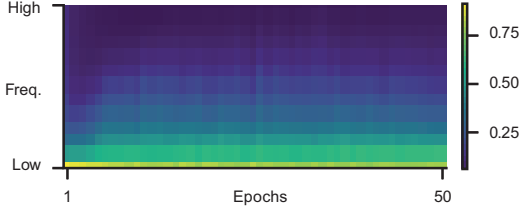

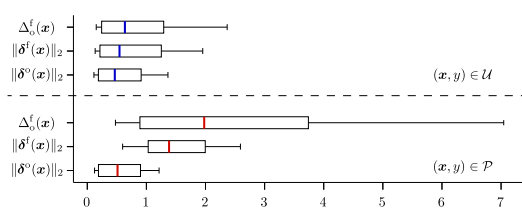

Figure 8: Energy of adversarial perturbations on subspaces of the DCT during adv. training of CIFAR-10 (DenseNet-121). Plot shows 95-perc.

Figure 9: Margin distribution in different directions of a ResNet-18 trained on CIFAR-10 and fine-tuned on 100 DeepFool examples.

Overall, we see that adversarial training exploits the sensitivity of the network to small changes in the training samples to hide some discriminative features from the model. This is especially clear when we compare the CIFAR-10 values in Fig. 7b and Fig. 2b, where it becomes evident that some previously used discriminative features in the high frequencies are completely overlooked by the adversarially trained network. In the following example, we show that, in practice, it is not actually necessary to change the position of all training points to induce a large invariance reaction.

**Invariance can be triggered by just a few samples** Modifying the position of just a *minimal* number of training samples is enough to locally introduce excessive invariance on a classifier. To demonstrate this, we take a ResNet-18 (test: $90\%$) trained on CIFAR-10, and randomly select a set of 100 training samples $\mathcal{P} \subset \mathcal{T}$. We fine-tune this classifier replacing those 100 samples with $(\boldsymbol{x} + \boldsymbol{\delta}^{\mathrm{o}}(\boldsymbol{x}), y)$ in $\mathcal{P}$ (test: $90\%$), where $\boldsymbol{\delta}^{\mathrm{o}}$ and $\boldsymbol{\delta}^{\mathrm{f}}$ represent the adversarial perturbations for the original and fine-tuned network, respectively.

Fig. 9 shows the magnitude of these perturbations both for the 100 adversarially perturbed points $\mathcal{P} \subset \mathcal{T}$ and for a subset of $1,000$ unmodified samples $\mathcal{U} \subset \mathcal{T}$. Here, we can clearly see that, after fine-tuning, the boundaries around $\mathcal{P}$ have been completely modified, showing a large increase in the distance to the boundary in the direction of the original adversarial perturbation $\Delta_{\mathrm{o}}^{\mathrm{f}}(\boldsymbol{x})$ for $(\boldsymbol{x}, y) \in \mathcal{P}$. Meanwhile, the boundaries around $\mathcal{U}$ have not seen such a dramatic change.

This means that modifying the position of *only* a small fraction of the training samples can induce a large change in the shape of the boundary. Note that this dependency on a few samples resembles the one of support vector machines [37], whose decision boundaries are defined by the position of a few supporting vectors. However, in contrast to SVMs, deep neural networks are not guaranteed to maximize margin in the input space (see Fig. 6), and the points that support their boundaries need not be the ones closest to them, hence rendering their identification much harder.

## 5 Concluding remarks

In this paper, we proposed a new framework that permits to relate data features and margin along specific directions. We also explained how the inductive bias of the learning algorithm shapes the decision boundaries of neural networks by creating boundaries that are invariant to non-discriminative directions. We further showed that these boundaries are very sensitive to the exact position of the training samples, and that this enables adversarial training to build more robust classifiers.

**Future directions** We believe that our new framework can be used in future research to investigate the connections between training features and the macroscopic geometry of deep models. This can serve as a tool to obtain new insights on the intriguing properties of deep networks such as their catastrophic forgetting [34]. On the practical side, there are some important applications that could benefit from our findings. In terms of robustness, identifying the small subspace of discriminative features of a network can lead to faster black box-attacks by restricting the search space of the perturbations. In fact our analysis explains why in recent attacks [7, 21, 38] using low-frequency perturbations improves the query efficiency. Simultaneously, the dependency of boundaries to just a few training samples can be exploited to design faster adversarial training schemes, and is a clear avenue for future research in active learning [39]. Finally, having a better understanding about the mechanisms that lead to excessive invariance [14] after adversarial training could help boost the standard accuracy of robust models.

## Broader impact

In this work, we build on the mechanisms of adversarial machine learning and propose a new framework that connects the microscopic features of a dataset (i.e., position of the training samples in the input space) to the macroscopic properties of the learned models (e.g., distance to the decision boundary). Our methodology sheds light onto the inductive bias that deep classifiers exploit for shaping their decision boundaries and might explain the successes and limitations of deep learning.

Part of our work continues a recent line of research that shows that the way neural networks perceive the image spectrum is very different to the way humans do. In fact, based on the margin distributions for different frequencies that we measure, we can see that neural networks can sometimes use features in the higher end of the spectrum which are invisible to the human eye (see Fig. 2 and [4, 20]). A positive application of our work would therefore be the use of this knowledge and some methods derived from our experimental framework to better align the behavior of neural networks to the human visual system perception. This could have positive implications in the interpretability of neural networks when deployed on some domains where it is necessary to explain the decisions of a classifier, e.g., medical imaging.

We see the main possible negative implication of our work in the malicious exploitation of the discriminative features of the datasets for generating more advanced and efficient adversarial attacks. When deploying deep models into the real world, especially for safety-critical applications, it is of high importance that practitioners are aware of the low margin blindspots in the classifiers and make the best to protect them.

## Acknowledgments

We thank Maksym Andriushchenko, and Evangelos Alexiou for their fruitful discussions and feedback. This work has been partially supported by the CHIST-ERA program under Swiss NSF Grant 20CH21_180444, and partially by Google via a Postdoctoral Fellowship and a GCP Research Credit Award.

## Footnotes

[1]In general all details of our experiments are listed in the Supp. material.

[2]We do not enforce the $[0, 1]^D$ box constraints on the adversarial images, as we are not interested in finding "plausible" adversarial perturbations, but in measuring the distance to $\mathcal{B}(f)$.

[3]This is indeed a desired property for any classification method, but note that for neural networks the existence of adversarial examples contests the idea of it being a reasonable assumption.

[4]In Sec. A of Supp. material we provide a theoretical characterization of this effect for a linear classifier.

[5]Experiments on more CNNs (with similar findings) are presented in Sec. I of Supp. material.

[6]See Sec. I of Supp. material for a similar analysis including off-diagonal subspaces.

[7]We do not plot the full distribution to avoid clutter. The 5-perc. of the margin in the last subspace is $5.05$.

[8]A similar effect was shown on ImageNet [20], although the network was only tested on filtered data. For MNIST, training on a low-pass version (with bandwidth $14 \times 14$) yields no drop in test accuracy. This makes sense, as the original MNIST trained networks are mostly exploiting low frequencies and already have high margins in the high frequencies. There might exist discriminative information in the high frequencies, but the network does not exploit it (see Sec. C in the Supp. material).

[9]Note that this particular pattern, can in principle classify any dataset with $\rho = 20$, no matter the value of $\epsilon$.

[10]The analogous effect for the "flipped" datasets (cf. Section 3.2.1) is detailed in Sec. M of Supp. material.

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
