[Supplementary Material]

# Supplementary material: Hold me tight! Influence of discriminative features on deep network boundaries

## Contents

# A  Theoretical margin distribution of a linear classifier

In this section we prove that even for linear classifiers trained on $\mathcal{T}_1(\epsilon, \rho, N)$ the distribution of margins along non-discriminative directions will never be infinite, and that it will have a large variance (c.f. Section 3.1). This effect is due to the finiteness of the training set which boosts the influence of the non-discriminative directions in the final solution of the optimization. In particular, we show this for the linear classifier introduced in [1] and prove the following proposition:

**Proposition.** *Let $f(\boldsymbol{x}) = \boldsymbol{w}^T \boldsymbol{x}$ be a linear classifier trained on $\mathcal{T}_1(\epsilon, \sigma, N)$ using one-step gradient descent initialized with $\boldsymbol{w} = 0$ and $\alpha = 1$ to maximize $f(\boldsymbol{x}^{(i)})y^{(i)}$ for every sample, and let $\xi^2(\boldsymbol{x})$ denote the ratio between the margin in the direction of the discriminative feature $\mathrm{span}\{\boldsymbol{u}_1\}$ and the margin in an orthogonal random subspace $\mathcal{S}_{orth} \subseteq \mathrm{span}\{\boldsymbol{u}_1\}^\perp$ of dimension $|\mathcal{S}| = S \leq D - 1$, i.e.,*

$$\xi^2(\boldsymbol{x}) = \frac{\|\boldsymbol{\delta}_{\mathrm{span}\{\boldsymbol{u}_1\}}(\boldsymbol{x})\|_2^2}{\|\boldsymbol{\delta}_{\mathcal{S}_{orth}}(\boldsymbol{x})\|_2^2},$$

*The distribution of $\xi^2(\boldsymbol{x})$ is independent of $\boldsymbol{x}$ and follows $\xi^2(\boldsymbol{x}) \sim N\sigma^2 \chi_S^2$, where $\chi_S^2$ denotes the Chi-squared distribution with $S$ degrees of freedom. In particular,*

$$\mathrm{median}(\xi^2) = \mathcal{O}\left(\frac{\sigma^2}{N\epsilon^2}S\right) \quad \text{and} \quad \mathrm{Var}(\xi^2) = \frac{2\sigma^4}{N^2\epsilon^4}S$$

*Proof.* First, note that the weights of the classifier, after one step of GD, are

$$\boldsymbol{w} = \nabla_{\boldsymbol{w}} \sum_{i=0}^{N-1} f\left(\boldsymbol{x}^{(i)}\right) y^{(i)} = \sum_{i=0}^{N-1} \boldsymbol{x}^{(i)} y^{(i)} = \boldsymbol{U} \sum_{i=0}^{N-1} \left(\boldsymbol{x}_1^{(i)} \oplus \boldsymbol{x}_2^{(i)}\right) y^{(i)}.$$

Hence,

$$\boldsymbol{w} = \boldsymbol{U}(\boldsymbol{w}_1 \oplus \boldsymbol{w}_2) \quad \text{with} \quad \begin{cases} \boldsymbol{w}_1 = \sum_{i=0}^{N-1} y^{(i)} \boldsymbol{x}_1^{(i)} = N\epsilon \\ \boldsymbol{w}_2 = \sum_{i=0}^{N-1} y^{(i)} \boldsymbol{x}_2^{(i)} \end{cases}$$

Since $y^{(i)}$ are uniform discrete random variables taking values from $\{-1, +1\}$, $\boldsymbol{x}_2^{(i)}$ are standard normal random variables independent from $y^{(i)}$, it can be shown that their product $y^{(i)}\boldsymbol{x}_2^{(i)}$ is also a standard normal random variable. Hence, $\boldsymbol{w}_2 \sim \mathcal{N}(0, N\sigma^2 \boldsymbol{I}_{D-1})$.

Recall that for linear classifiers the distance to the decision boundary of a point $\boldsymbol{x}$ on a vector subspace $\mathcal{S} \subseteq \mathbb{R}^D$ can be computed in closed form as

$$\|\boldsymbol{\delta}_{\mathcal{S}}(\boldsymbol{x})\|_2 = \frac{|\boldsymbol{w}^T \boldsymbol{x}|}{\|\mathcal{P}_{\mathcal{S}}(\boldsymbol{w})\|_2}$$

where $\mathcal{P}_{\mathcal{S}} : \mathbb{R}^D \to \mathbb{R}^D$ denotes the orthogonal projection operator onto the subspace $\mathcal{S}$. Considering this, we can compute both the margin in $\mathrm{span}\{\boldsymbol{u}_1\}$ and $\mathcal{S}_{\mathrm{orth}}$ as

$$\|\boldsymbol{\delta}_{\mathrm{span}\{\boldsymbol{u}_1\}}(\boldsymbol{x})\|_2 = \frac{|\boldsymbol{w}^T \boldsymbol{x}|}{\|\mathcal{P}_{\mathrm{span}\{\boldsymbol{u}_1\}}(\boldsymbol{w})\|_2} = \frac{|\boldsymbol{w}^T \boldsymbol{x}|}{\|\boldsymbol{w}_1\|_2},$$

$$\|\boldsymbol{\delta}_{\mathcal{S}_{\mathrm{orth}}}(\boldsymbol{x})\|_2 = \frac{|\boldsymbol{w}^T \boldsymbol{x}|}{\|\mathcal{P}_{\mathcal{S}_{\mathrm{orth}}}(\boldsymbol{w})\|_2} = \frac{|\boldsymbol{w}^T \boldsymbol{x}|}{\|\mathcal{P}_{\mathcal{S}_{\mathrm{orth}}^{D-1}}(\boldsymbol{w}_2)\|_2},$$

where $\mathcal{S}_{\mathrm{orth}}^{D-1} \subseteq \mathbb{R}^{D-1}$ is the subspace generated by the last $D - 1$ components of the vectors in $\mathcal{S}$.

Squaring these distances and taking their ratio we have

$$\xi^2 = \frac{\|\boldsymbol{\delta}_{\mathrm{span}\{\boldsymbol{u}_1\}}(\boldsymbol{x})\|_2^2}{\|\boldsymbol{\delta}_{\mathcal{S}_{\mathrm{orth}}}(\boldsymbol{x})\|_2^2} = \frac{\|\mathcal{P}_{\mathcal{S}_{\mathrm{orth}}^{D-1}}(\boldsymbol{w}_2)\|_2^2}{\|\boldsymbol{w}_1\|_2^2}.$$

Note now that due to the rotational symmetry of $\mathcal{N}(0, \boldsymbol{I}_{D-1})$

$$\mathcal{P}_{\mathcal{S}_{\mathrm{orth}}^{D-1}}(\boldsymbol{w}_2) \sim \mathcal{N}\left(0, N\sigma^2 \boldsymbol{U}_{\mathcal{S}_{\mathrm{orth}}^{D-1}} \boldsymbol{I}_S \boldsymbol{U}_{\mathcal{S}_{\mathrm{orth}}^{D-1}}^T\right),$$

where $\boldsymbol{U}_{\mathcal{S}^{D-1}_{\text{orth}}} \in \mathbb{R}^{D-1 \times S}$ is a matrix whose columns form an orthonormal basis of $\mathcal{S}^{D-1}_{\text{orth}}$. Hence, $\|\mathcal{P}_{\mathcal{S}^{D-1}_{\text{orth}}}(\boldsymbol{w}_2)\|_2^2 \sim N\sigma^2 \chi^2_S$ and

$$\xi^2 = \frac{\|\mathcal{P}_{\mathcal{S}^{D-1}_{\text{orth}}}(\boldsymbol{w}_2)\|_2^2}{\|\boldsymbol{w}_1\|_2^2} = \frac{\|\mathcal{P}_{\mathcal{S}^{D-1}_{\text{orth}}}(\boldsymbol{w}_2)\|_2^2}{N^2\epsilon^2} \sim \frac{\sigma^2}{N\epsilon^2}\chi^2_S.$$

Finally, plugging in the expression for the median and variance of a Chi-squared distribution we get

$$\text{median}(\xi^2) \approx \frac{\sigma^2}{N\epsilon^2} S \left(1 - \frac{2}{9S}\right)^3,$$

and

$$\text{Var}(\xi^2) = \frac{2\sigma^4}{N^2\epsilon^4} S.$$

$\square$

Clearly, $\text{median}(\xi^2)$ decreases asymptotically with respect to the number of samples. Nevertheless, due to the finiteness of the training set, small but non-zero values of $\xi^2$ are unavoidable. Similarly, $\text{Var}(\xi^2)$ only decreases quadratically with the number of samples and grows linearly with the dimensionality of $\mathcal{S}_{\text{orth}}$. Hence, some fluctuations in the measured margins are expected even for linear classifiers.

We demonstrate this effect in practice by repeating the experiment of Sec. 3.1, where instead of an MLP we use a simple logistic regression (see Table S1).Clearly, although the values along $\text{span}\{\boldsymbol{u}_1\}^\perp$ are quite large, they are still finite. This demonstrates that due to the finiteness of the training set and its high-dimensionality the influence of the non-discriminative directions in the final solution is significant.

Table S1: Margin statistics of a logistic regressor trained on $\mathcal{T}_1(\epsilon = 5, \sigma = 1)$ along different directions ($N = 10,000$, $M = 1,000$, $S = 3$).

|          | $\boldsymbol{u}_1$ | $\text{span}\{\boldsymbol{u}_1\}^\perp$ | $\mathcal{S}_{\text{ORTH}}$ | $\mathcal{S}_{\text{RAND}}$ |
|----------|------|-------|---------|--------|
| 5-PERC.  | 2.39 | 36.7  | 184.95  | 11.57  |
| MEDIAN   | 2.49 | 38.3  | 192.98  | 12.08  |
| 95-PERC. | 2.60 | 39.92 | 201.16  | 12.59  |

# B  Examples of frequency "flipped" images

Figure S1 shows a few example images of the frequency "flipped" versions of the standard computer vision datasets.

(a) ImageNet

(b) CIFAR-10

(c) MNIST

Figure S1: "Flipped" image examples. **Top** rows show original images and **bottom** rows the "flipped" versions.

# C  Invariance and elasticity on MNIST data

We further validate our observation of Section 3.2.2 that small margin do indeed corresponds to directions containing discriminative features in the training set, but this time for a different dataset (MNIST), on a different network (ResNet-18), and using different discriminative features (high-frequency). In particular, we create a high-pass filtered version of MNIST (MNIST$_{HP}$), where we completely remove the frequency components in a $14 \times 14$ square at the top left of the diagonal of the DCT-transformed images. This way we ensure that every pairwise connection between the training images (features) has zero components outside of this frequency subspace. The margin distribution of $1,000$ MNIST test samples for a ResNet-18 trained on MNIST$_{HP}$ is illustrated in Figure S2. Indeed, similarly to the observations on CIFAR-10, by eliminating the low frequency features, we have forced an increased margin along these directions, while forcing the network to focus on the previously unused high frequency features.

Figure S2: Median margin of test samples from MNIST for a ResNet-18 trained on MNIST$_{HP}$ from scratch (test: $98.71\%$).

# D   Connections to catastrophic forgetting

The elasticity to the modification of features during training gives a new perspective to the theory of catastrophic forgetting [2], as it confirms that the decision boundaries of a neural network can only exist for as long as the classifier is trained with the samples (features) that hold them together. In particular, we demonstrate this by adding and removing points from a dataset such that its discriminative features are modified during training, and hence artificially causing an elastic response on the network.

To this end, we train a DenseNet-121 on a new dataset $\mathcal{T}_{LP \cup HP} = \mathcal{T}_{LP} \cup \mathcal{T}_{HP}$ formed by the union of two filtered variants of CIFAR-10: $\mathcal{T}_{LP}$ is constructed by retaining only the frequency components in a $16 \times 16$ square at the top-left of of the DCT-transformed CIFAR-10 images (low-pass), while for $\mathcal{T}_{HP}$ only the frequency components in a $16 \times 16$ square at the bottom-right of the DCT (high-pass). This classifier has a test accuracy of $86.59\%$ and $57.29\%$ on $\mathcal{T}_{LP}$ and $\mathcal{T}_{HP}$, respectively. The median margin of $1,000$ $\mathcal{T}_{LP}$ test samples along different frequencies for this classifier is shown in blue in Figure S3. As expected, the classifier has picked features across the whole spectrum with the low frequency ones probably belonging to boundaries separating samples in $\mathcal{T}_{LP}$, and the high frequency ones separating samples from $\mathcal{T}_{LP}$ and $\mathcal{T}_{HP}$[1].

(a) Zoom-out axes for observing the general invariance.   (b) Zoom-in axes for a more detailed observation.

Figure S3: Median margin of $\mathcal{T}_{LP}$ test samples for a DenseNet-121. **Blue:** trained on $\mathcal{T}_{LP \cup HP}$; **Red:** after forgetting $\mathcal{T}_{HP}$; **Green:** after recovering $\mathcal{T}_{HP}$.

After this, we continue training the network with a linearly decaying learning rate (max. $\alpha = 0.05$) for another 30 epochs, but using only $\mathcal{T}_{LP}$, achieving a final test accuracy of $87.81\%$ and $10.01\%$ on $\mathcal{T}_{LP}$ and $\mathcal{T}_{HP}$, respectively. Again, Figure S3 shows in red the median margin along different frequencies on test samples from $\mathcal{T}_{LP}$. The new median margin is clearly invariant on the high frequencies – where $\mathcal{T}_{LP}$ has no discriminative features – and the classifier has completely *erased* the boundaries that it previously had in these regions, regardless of the fact that those boundaries did not harm the classification accuracy on $\mathcal{T}_{LP}$.

Finally, we investigate if the network is able to recover the forgotten decision boundaries that were used to classify $\mathcal{T}_{HP}$. We continue training the network ("forgotten" $\mathcal{T}_{HP}$) for another 30 epochs, but this time by using the whole $\mathcal{T}_{LP \cup HP}$. Now this classifier achieves a final test accuracy of $86.1\%$ and $59.11\%$ on $\mathcal{T}_{LP}$ and $\mathcal{T}_{HP}$ respectively, which are very close to the corresponding accuracies of the initial network trained from scratch on $\mathcal{T}_{LP \cup HP}$ (recall: $86.59\%$ and $57.29\%$). The new median margin for this classifier is shown in green in Figure S3. As we can see by comparing the green to the blue curve, the decision boundaries along the high-frequency directions can be recovered quite successfully.

# E  Examples of filtered images

Figure S4 shows a few example images of the filtered versions of the standard computer vision datasets used in the Section 3.2.2, C and D.

(a) CIFAR-10 (**Top** original images, **middle** low-pass and **bottom** high-pass)

(b) MNIST (**Top** original images and **bottom** high-pass)

Figure S4: Filtered image examples.

# F  Subspace sampling of the DCT

In most of our experiments with real data we measured the margin of $M$ samples on a sequence of subspaces created using blocks from the DCT. In particular, we use a sequence of $K \times K$ blocks sampled from the DCT tensor either from a sliding window on the diagonal with step size $T$ or a grid with stride $T$ (c.f. Figure S5).

Figure S5: Diagram illustrating the main parameters defining the subspace sequence from the diagonal of the DCT.

# G    Training parameters

Table S2 shows the performance and training parameters of the different networks used in the paper. Note that the hyperparameters of these networks were not optimized in any form during this work. Instead they were selected from a set of best practices from the DAWNBench submissions that have been empirically shown to give a good trade-off in terms of convergence speed and performance. In this sense, especially for the non-standard datasets (e.g., "flipped" datasets), the final performance might not be the best reflection of the highest achievable performance of a given architecture. In fact, since the goal of our experiments is not to achieve the most robust models on such non-standard datasets, but rather investigate how the previously observed trends are represented in these new classifiers, no further hyperparameter tuning was applied.

Table S2: Performance and training parameters of multiple networks trained on different datasets. All networks have been trained using SGD with momentum $0.9$ and a weight decay of $5 \times 10^{-4}$. For ImageNet, the training parameters are not known, since we use the pretrained models from PyTorch. For "flipped" ImageNet, the weight decay was set to $10^{-4}$, while for computational reasons the training was executed until the $68^{\text{th}}$ epoch.

| DATASET | NETWORK | TEST ACC. | EPOCHS | LR SCHEDULE | MAX. LR | BATCH |
|---|---|---|---|---|---|---|
| MNIST | LENET<br>RESNET-18 | 99.35%<br>99.53% | 30 | TRIANG. | 0.21 | 128 |
| MNIST FLIPPED | LENET<br>RESNET-18 | 99.34%<br>99.52% | 30 | TRIANG. | 0.21 | 128 |
| CIFAR-10 | VGG-19<br>RESNET-18<br>DENSENET-121 | 89.39%<br>90.05%<br>93.03% | 50 | TRIANG. | 0.21 | 128 |
| CIFAR-10 LOW PASS | VGG-19<br>RESNET-18<br>DENSENET-121 | 84.81%<br>84.77%<br>88.51% | 50 | TRIANG. | 0.21 | 128 |
| CIFAR-10 FLIPPED | VGG-19<br>RESNET-18<br>DENSENET-121 | 87.42%<br>88.67%<br>91.19% | 50 | TRIANG. | 0.21 | 128 |
| IMAGENET | VGG-16<br>RESNET-50<br>DENSENET-121 | 71.59%<br>76.15%<br>74.65% | – | – | – | – |
| IMAGENET FLIPPED | RESNET-50 | 68.12% | 90(68) | PIECEWISE CONSTANT | 0.1 | 256 |

As mentioned in the paper, all the experiments with synthetic data were trained in the same way, namely using SGD with a linearly decaying learning rate (max lr. 0.1), no explicit regularization, and trained for 500 epochs.

# H Cross-dataset performance

We now show the performance of different networks trained with different variants of the standard computer vision datasets and tested on the rest.

Table S3: Multiple networks trained on a specific version of MNIST, but evaluated on different variations of it. Rows denote the dataset that each network is trained on, and columns the dataset they are evaluated on. Values on the diagonal correspond to the same variation.

|  |  | MNIST | MNIST FLIPPED | MNIST HIGH PASS |
|---|---|---|---|---|
| MNIST | LENET | 99.35% | 18.73% | 44.09% |
|  | RESNET-18 | 99.53% | 11.88% | 15.73% |
| MNIST FLIPPED | LENET | 10.52% | 99.34% | 9.87% |
|  | RESNET-18 | 16.59% | 99.52% | 11.23% |
| MNIST HIGH PASS | LENET | 96.35% | 42.36% | 98.65% |
|  | RESNET-18 | 88.38% | 21.48% | 98.71% |

Table S4: Multiple networks trained on a specific version of CIFAR-10, but evaluated on different variations of it. Rows denote the dataset that each network is trained on, and columns the dataset they are evaluated on. Values on the diagonal correspond to the same variation.

|  |  | CIFAR-10 | CIFAR-10 FLIPPED | CIFAR-10 LOW PASS |
|---|---|---|---|---|
| CIFAR-10 | VGG-19 | 89.39% | 10.63% | 61.4% |
|  | RESNET-18 | 90.05% | 10% | 46.99% |
|  | DENSENET-121 | 93.03% | 10.3% | 27.45% |
| CIFAR-10 FLIPPED | VGG-19 | 10.77% | 87.42% | 10.79% |
|  | RESNET-18 | 9.91% | 88.67% | 9.97% |
|  | DENSENET-121 | 9.98% | 91.19% | 10% |
| CIFAR-10 LOW PASS | VGG-19 | 85.16% | 10.52% | 84.81% |
|  | RESNET-18 | 85.47% | 10.45% | 84.77% |
|  | DENSENET-121 | 89.67% | 10.45% | 88.51% |

Table S5: Multiple networks trained on a specific version of ImageNet, but evaluated on different variations of it. Rows denote the dataset that each network is trained on, and columns the dataset they are evaluated on. Values on the diagonal correspond to the same variation.

|  |  | IMAGENET | IMAGENET FLIPPED |
|---|---|---|---|
| IMAGENET | VGG-16 | 71.59% | 0.106% |
|  | RESNET-50 | 76.15% | 0.292% |
|  | DENSENET-121 | 74.65% | 0.22% |
| IMAGENET FLIPPED | RESNET-50 | 0.184% | 68.12% |

# I  Margin distribution for standard networks

We show here the margin distribution on the diagonal of the DCT for different networks trained using multiple datasets using the setup specified in Section G. We also show the median margin for the same $M$ samples on a grid from the DCT.

The first thing to notice is that, for a given dataset, the trend of the margins are quite similar regardless the network architecture. Also, regardless the evaluation (diagonal or grid), the observed margins between train and test samples are very similar, with the differences in the values being quite minimal. Furthermore, for the grid evaluations, the trend of the median margins with respect to subspaces of different frequencies (increasing from low to high frequencies) is similar to the corresponding one of the diagonal evaluations. Hence, the choice of the diagonal of the DCT is sufficient for measuring the margin along directions of the frequency spectrum. Finally, in every evaluation (diagonal or grid) and for every data set (train or test), "flipping" the representation of the data results in "flipped" margins as well, with CIFAR-10 results being an exception due to the quite uniform distribution of the margin across the whole frequency spectrum.

## I.1  MNIST

(a) LeNet (Test)    (b) ResNet-18 (Test)

(c) LeNet (Train)    (d) ResNet-18 (Train)

Figure S6: Diagonal **MNIST** ($M = 1,000, K = 8, T = 1$)

(a) LeNet (Test)    (b) ResNet-18 (Test)

(c) LeNet (Train)    (d) ResNet-18 (Train)

Figure S7: Grid **MNIST** ($M = 500, K = 8, T = 3$)

## I.2 MNIST "flipped"

(a) LeNet (Test)   (b) ResNet-18 (Test)

(c) LeNet (Train)   (d) ResNet-18 (Train)

Figure S8: Diagonal **MNIST "flipped"** ($M = 1,000, K = 8, T = 1$)

(a) LeNet (Test)   (b) ResNet-18 (Test)

(c) LeNet (Train)   (d) ResNet-18 (Train)

Figure S9: Grid **MNIST "flipped"** ($M = 500, K = 8, T = 3$)

## I.3 CIFAR-10

(a) VGG-16 (Test)   (b) ResNet-18 (Test)   (c) DenseNet-121 (Test)

(d) VGG-16 (Train)   (e) ResNet-18 (Train)   (f) DenseNet-121 (Train)

Figure S10: Diagonal **CIFAR-10** ($M = 1,000, K = 8, T = 2$)

(a) VGG-19 (Test)  (b) ResNet-18 (Test)  (c) DenseNet-121 (Test)

(d) VGG-19 (Train)  (e) ResNet-18 (Train)  (f) DenseNet-121 (Train)

Figure S11: Grid **CIFAR-10** ($M = 500, K = 8, T = 4$)

## I.4 CIFAR-10 "flipped"

(a) VGG-16 (Test)  (b) ResNet-18 (Test)  (c) DenseNet-121 (Test)

(d) VGG-16 (Train)  (e) ResNet-18 (Train)  (f) DenseNet-121 (Train)

Figure S12: Diagonal **CIFAR-10 "flipped"** ($M = 1,000, K = 8, T = 2$)

(a) VGG-19 (Test)  (b) ResNet-18 (Test)  (c) DenseNet-121 (Test)

(d) VGG-19 (Train)  (e) ResNet-18 (Train)  (f) DenseNet-121 (Train)

Figure S13: Grid **CIFAR-10 "flipped"** ($M = 500, K = 8, T = 4$)

## I.5 ImageNet

(a) VGG-16 (Test)  (b) ResNet-50 (Test)  (c) DenseNet-121 (Test)

(d) VGG-16 (Train)  (e) ResNet-50 (Train)  (f) DenseNet-121 (Train)

Figure S14: Diagonal **ImageNet** ($M = 500$, $K = 16$, $T = 16$)

(a) VGG-16 (Test)  (b) ResNet-18 (Test)  (c) DenseNet-121 (Test)

(d) VGG-16 (Train)  (e) ResNet-50 (Train)  (f) DenseNet-121 (Train)

Figure S15: Grid **ImageNet** ($M = 250$, $K = 16$, $T = 28$)

## I.6 ImageNet "flipped"

(a) ResNet-50 (Test)    (b) ResNet-50 (Train)

Figure S16: Diagonal **ImageNet "flipped"** ($M = 500, K = 16, T = 16$)

(a) ResNet-50 (Test)    (b) ResNet-50 (Train)

Figure S17: Grid **ImageNet "flipped"** ($M = 250, K = 16, T = 28$)

## J  Adversarial training parameters

Table S6 shows the performance and adversarial training parameters of the different networks used in the paper. Note that the hyperparameters of these networks were not optimized in any form during this work. Instead they were selected from a set of best practices from the DAWNBench submissions that have been empirically shown to give a good trade-off in terms of convergence speed and performance. Again, as stated in Section G, especially for the non-standard datasets (e.g., "flipped" datasets), the final performance might not be the best reflection of the highest achievable performance or robustness of a given architecture, since no further hyperparameter tuning was applied.

Table S6: Performance and attack parameters of multiple networks adversarially trained using $\ell_2$-PGD. The training parameters are similar to the ones of Table S2. For ImageNet we use the adversarially trained ResNet-50 provided by [3].

| DATASET | NETWORK | STANDARD TEST ACC. | ADV. TEST ACC. | EPOCHS | $\ell_2$ BALL RADIUS | STEPS |
|---------|---------|--------------------|----------------|--------|----------------------|-------|
| MNIST | LENET RESNET-18 | 98.32% 98.89% | 76.01% 80.26% | 25 | 2 | 7 |
| MNIST FLIPPED | LENET RESNET-18 | 98.29% 98.75% | 74.68% 81.97% | 25 | 2 | 7 |
| CIFAR-10 | VGG-19 RESNET-18 DENSENET-121 | 73.76% 82.20% 82.90% | 50.15% 52.38% 54.86% | 50 | 1 | 7 |
| CIFAR-10 FLIPPED | VGG-19 RESNET-18 DENSENET-121 | 71.39% 73.64% 78.32% | 35.64% 37.24% 42.32% | 50 | 1 | 7 |
| IMAGENET | RESNET-50 | 57.90% | 35.16 | – | 3 | 20 |

# K   Description of L2-PGD attack on frequency "flipped" data

Adversarial training [4] is the de-facto method used to improve the robustness of modern deep classifiers. It consists in the approximation of the robust classification problem $\min_f \max_{\boldsymbol{\delta} \in \mathcal{C}} \mathcal{L}(f(\boldsymbol{x} + \boldsymbol{\delta}))$ with an alternating algorithm that solves the outer maximization using a variant of stochastic gradient descent, and the inner maximization using some adversarial attack (e.g., PGD). The constraint set $\mathcal{C} \subseteq \mathbb{R}^D$ encodes the "imperceptibility" of the perturbation.

In our case, when dealing with natural images coming from the standard datasets (i.e., MNIST, CIFAR-10 and ImageNet) we use the standard $\ell_2$ PGD attack to approximate the inner maximization. This attack consists in the solution of $\arg\max_{\boldsymbol{\delta} \in \mathcal{C}} \mathcal{L}(f(\boldsymbol{x} + \boldsymbol{\delta}))$ using projected steepest descent, i.e., iterating

$$\boldsymbol{\delta}_{n+1} = \mathcal{P}_{\mathcal{C}}\left(\boldsymbol{\delta}_n + \alpha \frac{\nabla_{\boldsymbol{\delta}}\mathcal{L}(f(\boldsymbol{x} + \boldsymbol{\delta}_n))}{\|\nabla_{\boldsymbol{\delta}}\mathcal{L}(f(\boldsymbol{x} + \boldsymbol{\delta}_n))\|_2}\right),$$

where $\mathcal{C} = \{\boldsymbol{\delta} \in \mathbb{R}^D : \|\boldsymbol{\delta}\|_2^2 \leq \epsilon, \quad \boldsymbol{0} \preceq \boldsymbol{\delta} \preceq \boldsymbol{1}\}$. The projection operator $\mathcal{P}_{\mathcal{C}} : \mathbb{R}^D \to \mathbb{R}^D$ can efficiently be implemented as

$$\mathcal{P}_{\mathcal{C}}(\boldsymbol{x}) = \text{clip}_{[0,1]}\left(\min\{\|\boldsymbol{\delta}\|_2, \epsilon\}\frac{\boldsymbol{\delta}}{\|\boldsymbol{\delta}\|_2}\right),$$

where

$$[\text{clip}_{[0,1]}(\boldsymbol{x})]_i = \begin{cases} 0 & [\boldsymbol{x}]_i \leq 0 \\ [\boldsymbol{x}]_i & [\boldsymbol{x}]_i < 0 \leq 1 \\ 1 & [\boldsymbol{x}]_i > 1 \end{cases}.$$

However, when we train using "flipped" data we need to make sure that we also transform the constraint set $\mathcal{C}$. Indeed, recall that the goal of training with "flipped" datasets is to check that the margin distribution approximately follows the data representation. Adversarial training tries to maximize the loss of the classifier by finding a worst-case example inside a constrained search space that is parameterized in terms of some properties of the input data (e.g., distance to a sample, or color box constraints). For this reason, if our goal is to check what happens when we only change the data representation but keep the same training scheme, it is important to make sure that adversarial training has the same search space regardless of the data representation. The flipping operator is reversible, which means we can always go back to our initial representation. Hence, by respecting the constraints over the initial representation, we make sure that the resulted adversarial examples in the new representation will still satisfy the constraints when reversed to the initial representation (image space). We achieve this reparameterization efficiently by modifying the projection operator on PGD.

Let $\hat{\boldsymbol{x}} = \boldsymbol{D}_{\text{DCT}}^T \text{flip}(\boldsymbol{D}_{\text{DCT}}\boldsymbol{x})$ denote a frequency "flipped" data sample. The $\ell_2$ PGD attack on this representation solves $\arg\max_{\hat{\boldsymbol{\delta}} \in \hat{\mathcal{C}}} \mathcal{L}(f(\hat{\boldsymbol{x}} + \hat{\boldsymbol{\delta}}))$, where $\hat{\mathcal{C}} = \left\{\hat{\boldsymbol{\delta}} \in \mathbb{R}^D : \boldsymbol{D}_{\text{DCT}}^T \text{flip}\left(\boldsymbol{D}_{\text{DCT}}\hat{\boldsymbol{\delta}}\right) \in \mathcal{C}\right\}$. Therefore, the new "flipped" PGD algorithm becomes

$$\hat{\boldsymbol{\delta}}_{n+1} = \mathcal{P}_{\hat{\mathcal{C}}}\left(\hat{\boldsymbol{\delta}}_n + \alpha \frac{\nabla_{\hat{\boldsymbol{\delta}}}\mathcal{L}(f(\hat{\boldsymbol{x}} + \hat{\boldsymbol{\delta}}_n))}{\|\nabla_{\hat{\boldsymbol{\delta}}}\mathcal{L}(f(\hat{\boldsymbol{x}} + \hat{\boldsymbol{\delta}}_n))\|_2}\right),$$

where $\mathcal{P}_{\hat{\mathcal{C}}}$ can be efficiently implemented using Dykstra's projection algorithm [5]. This is, start with $\hat{\boldsymbol{x}}_0 = \hat{\boldsymbol{x}}, \hat{\boldsymbol{p}}_0 = \hat{\boldsymbol{q}}_0 = \boldsymbol{0}$ and update by

$$\hat{\boldsymbol{y}}_k = \min\{\|\hat{\boldsymbol{x}}_k + \hat{\boldsymbol{p}}_k\|_2, \epsilon\} \ \frac{\hat{\boldsymbol{x}}_k + \hat{\boldsymbol{p}}_k}{\|\hat{\boldsymbol{x}}_k + \hat{\boldsymbol{p}}_k\|_2}$$

$$\hat{\boldsymbol{p}}_{k+1} = \hat{\boldsymbol{x}}_k + \hat{\boldsymbol{p}}_k - \hat{\boldsymbol{y}}_k$$

$$\boldsymbol{x}_{k+1} = \text{clip}_{[0,1]}\left(\boldsymbol{D}_{\text{DCT}}^T \text{flip}\left(\boldsymbol{D}_{\text{DCT}}(\hat{\boldsymbol{y}}_k + \hat{\boldsymbol{q}}_k)\right)\right)$$

$$\hat{\boldsymbol{x}}_{k+1} = \boldsymbol{D}_{\text{DCT}}^T \text{flip}\left(\boldsymbol{D}_{\text{DCT}}\boldsymbol{x}_{k+1}\right)$$

$$\hat{\boldsymbol{q}}_{k+1} = \hat{\boldsymbol{y}}_k + \hat{\boldsymbol{q}}_k - \hat{\boldsymbol{x}}_{k+1}.$$

The sequence $(\hat{\boldsymbol{x}}_k)$ converges to $\mathcal{P}_{\hat{\mathcal{C}}}(\hat{\boldsymbol{x}})$. In our experiments we use 5 iterations of the algorithm as these are enough to achieve a small projection error.

# L  Spectral decomposition on frequency "flipped" data

Following the results presented in Section 4.2, we now show in Figure S18 the spectral decomposition of the adversarial perturbations crafted during adversarial training for the frequency "flipped" CIFAR-10 dataset on a DenseNet-121 network. In contrast to the spectral decomposition of the perturbations on CIFAR-10 (left), the energy of the frequency "flipped" CIFAR-10 perturbations (right) remains concentrated in the high part of the spectrum during the whole training process, and has hardly any presence in the low frequencies. In other words, the frequency content of the $\ell_2$-PGD adversarial perturbations also "flips" (c.f. Section K and M).

(a) CIFAR-10 adversarially trained model.

(b) Frequency "flipped" CIFAR-10 adversarially trained model.

Figure S18: Energy decomposition in subspaces of the DCT diagonal of adversarial perturbations used during adversarial training ($\ell_2$ PGD with $\epsilon = 1$) on 1,000 (a) CIFAR-10 and (b) frequency "flipped" CIFAR-10 training samples per epoch for a DenseNet-121. The plot shows 95-percentile of energy.

# M  Margin distribution for adversarially trained networks

We show here the margin distribution on the diagonal of the DCT for different adversarially trained networks on multiple datasets using the setup specified in Section J. We also show the median margin for the same $M$ samples on a grid from the DCT.

The first thing to notice for the standard datasets is that, for every network and dataset, there is a huge increase along the high-frequency directions, when compared to the margins observed in Section I. Apart from these, similarly to the observations of Section I, the margins on both train and test samples are very similar, with the differences in the values being quite minimal, while again the trend of the margins with respect to subspaces of different frequencies (increasing from low to high frequencies) is similar in both the grid and the diagonal evaluations. Finally, in every evaluation (diagonal or grid) and for every data set (train or test), "flipping" the representation of the data results in "flipped" margins as well; even for the case of CIFAR-10 where for standard training (Figure S12) the "flipping" was not obvious due to the quite uniform distribution of the margin.

## M.1  MNIST

(a) LeNet (Test)

(b) ResNet-18 (Test)

(c) LeNet (Train)

(d) ResNet-18 (Train)

Figure S19: Diagonal **MNIST** adversarially trained ($M = 1,000, K = 8, T = 1$)

(a) LeNet (Test)

(b) ResNet-18 (Test)

(c) LeNet (Train)

(d) ResNet-18 (Train)

Figure S20: Grid **MNIST** adversarially trained ($M = 500, K = 8, T = 3$)

## M.2   MNIST "flipped"

(a) LeNet (Test)

(b) ResNet-18 (Test)

(c) LeNet (Train)

(d) ResNet-18 (Train)

Figure S21: Diagonal **MNIST "flipped"** adversarially trained ($M = 1,000, K = 8, T = 1$)

(a) LeNet (Test)

(b) ResNet-18 (Test)

(c) LeNet (Train)

(d) ResNet-18 (Train)

Figure S22: Grid **MNIST "flipped"** adversarially trained ($M = 500, K = 8, T = 3$)

## M.3   CIFAR-10

(a) VGG-16 (Test)

(b) ResNet-18 (Test)

(c) DenseNet-121 (Test)

(d) VGG-16 (Train)

(e) ResNet-18 (Train)

(f) DenseNet-121 (Train)

Figure S23: Diagonal **CIFAR-10** adversarially trained ($M = 1,000, K = 8, T = 2$)

(a) VGG-19 (Test)  (b) ResNet-18 (Test)  (c) DenseNet-121 (Test)

(d) VGG-19 (Train)  (e) ResNet-18 (Train)  (f) DenseNet-121 (Train)

Figure S24: Grid **CIFAR-10** adversarially trained ($M = 500$, $K = 8$, $T = 4$)

## M.4  CIFAR-10 "flipped"

(a) VGG-16 (Test)  (b) ResNet-18 (Test)  (c) DenseNet-121 (Test)

(d) VGG-16 (Train)  (e) ResNet-18 (Train)  (f) DenseNet-121 (Train)

Figure S25: Diagonal **CIFAR-10 "flipped"** adversarially trained ($M = 1,000$, $K = 8$, $T = 2$)

(a) VGG-19 (Test)  (b) ResNet-18 (Test)  (c) DenseNet-121 (Test)

(d) VGG-19 (Train)  (e) ResNet-18 (Train)  (f) DenseNet-121 (Train)

Figure S26: Grid **CIFAR-10 "flipped"** adversarially trained ($M = 500$, $K = 8$, $T = 4$)

## M.5 ImageNet

(a) ResNet-50 (Test)  (b) ResNet-50 (Train)

Figure S27: Diagonal **ImageNet** adversarially trained ($M = 500, K = 16, T = 16$)

(a) ResNet-50 (Test)  (b) ResNet-50 (Train)

Figure S28: Grid **ImageNet** adversarially trained ($M = 250, K = 16, T = 28$)

## N    Margin distribution on random subspaces

Finally we show the same evaluation of Section I performed using a random orthonormal basis instead of the DCT basis to demonstrate that the choice of basis is indeed important to identify the discriminative and non-discriminative directions of a network. Indeed, from Figure S29 it is clear that a random basis is not valid for this task as the margin in any random subspace is of the same order with high probability [6].

(a) MNIST (Test: 99.35%)  (b) CIFAR-10 (Test: 93.03%)

(c) MNIST flipped (Test: 99.34%)  (d) CIFAR-10 flipped (Test: 91.19%)

Figure S29: Margin distribution of test samples in subspaces taken from a random orthonormal matrix arranged as a tensor of the same dimensionality as the DCT tensor. Subspaces are taken from the diagonal with the same parameters as before. **Top**: (a) MNIST (LeNet), (b) CIFAR-10 (DenseNet-121) **Bottom**: (d) MNIST (LeNet) and (e) CIFAR-10 (DenseNet-121) trained on frequency "flipped" versions of the standard datasets.

## Footnotes

[1]$\mathcal{T}_{LP}$ and $\mathcal{T}_{HP}$ have only discriminative features in the low-frequency and high-frequency part of the spectrum, respectively.