[Reviews · NeurIPS 2020]

Review 1

Summary and Contributions: This paper proposes a new perspective that relates dataset features to the distance of samples to the decision boundary. The contributions are as follows: 1. They provide a new perspective on the relationship between the distance of a set of samples to the boundary, and the discriminative features used by a network. 2. They rigorously confirm the “common belief” that CNNs tend to behave as ideal classifiers and are approximately invariant to non-discriminative features of a dataset. 3. They further show that the construction of the decision boundary is extremely sensitive to the position of the training samples, such that very small perturbations in certain directions can utterly change the decision boundaries in some orthogonal directions. 4. They demonstrate that adversarial training exploits this training sensitivity and invariance bias to build robust classifiers.

Strengths: 1. They provide a new perspective on the relationship between the distance of a set of samples to the boundary, and the discriminative features used by a network. 2. They rigorously confirm the “common belief” that CNNs tend to behave as ideal classifiers and are approximately invariant to non-discriminative features of a dataset. 3. They further show that the construction of the decision boundary is extremely sensitive to the position of the training samples, such that very small perturbations in certain directions can utterly change the decision boundaries in some orthogonal directions. 4. They demonstrate that adversarial training exploits this training sensitivity and invariance bias to build robust classifiers.

Weaknesses: The main limitation is that the directions of features are only based on 2D-DCT in real applications. It would be great to show similar results for other types of directions for features.

Correctness: Great.

Clarity: Great.

Relation to Prior Work: Great.

Reproducibility: Yes

Additional Feedback: Update: After reading the response, I want to keep my scores.


Review 2

Summary and Contributions: The paper investigates the relationship between the distance of samples from the decision boundary and feature characteristics. To this end, the authors constrain the perturbations to a subregion of the space and then analyze the local decision boundary around a few samples using the margin computed from the norm of the perturbations.

Strengths: The analysis presented is interesting which gives further evidence neural networks exhibit invariance to non-discriminative features and that adversarial training where samples with slightly perturbed original samples change the geometry of the decision boundary. Specifically, perturbations in a specific direction are shown to render the classifier invariant in orthogonal directions and small margin perturbation is discriminative directions. Overall the paper is well written and easy to follow. The experiments are convincing sheds new light on the robustness of neural networks to perturbation directions based on frequency content.

Weaknesses: It is not clear why Cifar10 behaves differently compared to MNIST and Imagenet in terms of margin to perturbations. Could it be due to the blurriness of the Cifar10 images? The experiments with a low pass version of Cifar10 is interesting, how would similar analysis look like for the other datasets, would we see a much smaller drop in accuracy than the 3% observed for Cifar10? In the experiment in sec 4.2 will we see similar results if we constraint the adversarial training such that there are no high-frequency perturbations seen during training?

Correctness: They look correct.

Clarity: Yes the paper is easy to follow.

Relation to Prior Work: Yes, relevant work is mentioned and contributions are clearly mentioned.

Reproducibility: Yes

Additional Feedback: Typo Fig 1: top and bottom should be left right? Line 274: what is seen Update: Having read the other reviews and the rebuttal, I am inclined to maintain my score. I agree that the presentation can be improved and some choices can be better explianed to rule our potential confounders, but I still find the emperical findings interesting.


Review 3

Summary and Contributions: This paper leverages adversarial perturbations, using a subspace-constrained version of DeepFool, to quantify the margin. With this framework, on synthetic and real datasets, the authors relate the margin to discriminative vs non-discriminative features, the impact of perturbations on the network geometry, and how this provides an explanation for the success of adversarial training in removing features with small margin to increase the network’s robustness.

Strengths: This paper has a multitude of insightful experiments that clearly deliver a significant amount of information at the reader. The authors relate the margin perspective to the nature of catastrophic forgetting, showing that the decision boundaries of a neural network can only exist for as long as the classifier is trained with the features that hold them together. The authors use this framework to lend evidence on the success in exploiting low frequency perturbations, demonstrating that networks trained on MNIST and ImageNet present a strong invariance along high frequency directions and small margin along low frequency ones, showing that this is related to the fact that these networks mainly exploit discriminative features in the low frequencies of these datasets. Furthermore, the authors show that the energy of the perturbations during training is always concentrated in the low frequencies, however, the greatest effect on margin is seen on the orthogonal high frequency directions, hiding the high frequency discriminative features. This provides insight on why previous work in low frequency adversarial attacks have been successful even at targeting adversarially trained networks, when there is no explicit frequency bias within the framework of adversarial training.

Weaknesses: Given the large number of contributions, the paper was hard to follow for two main reasons. Firstly, many decisions were made without explanation. There are numerous examples of this within the work, in the choice of Deepfool, in the structure of the synthetic problem, in the choice of leveraging the DCT and thus focusing on the frequency spectrum of the data. Secondly, the paper lacked flow, it did not read as if subsequent sections naturally follow the previous one. It read as a list of experiments that are connected as they all utilize the proposed margin framework, but are not significantly related to each other beyond that, with the reason that an experiment precedes or follows a distinct experiment is unclear. Though the paper does make significant contributions to understanding the explored phenomena, it is difficult to generalize the contribution beyond a number of experiments with unexplained setting specifications which each utilize the margin framework to glean light on different questions. It would be great if all choices in the paper were motivated sufficiently, and if the paper provided a more unified takeaway, where the series of experiments naturally build on each other to come to that given conclusion.

Correctness: There does not appear to be any clear incorrect statements, though this may be obfuscated due to the lack of explanation given for particular design choices. The authors however do not appear to overclaim, suggesting their specified experimental setting generalizes significantly beyond what has been shown, so I do not see any blatantly incorrect notions.

Clarity: The paper is well written in the sense that all details are explained, but not well written in the sense that motivation and flow are missing, making it difficult to grasp a single core takeaway from this work.

Relation to Prior Work: it is discussed how this work differs from previous contributions, though the claims made in what has been contributed when relating to previous contributions are not completely clear to the reader, due to the the lack of clarity in what exactly can be taken away from this work.

Reproducibility: Yes

Additional Feedback: I'd welcome the authors to provide intuition for each of the design choices made in the construction of their experiments, and give their perspective on what conclusion should be obtained from the composition of each of their presented experiments, precisely detailing how each experiment is unique in its information contribution and required in order to argue that their conclusion is sufficiently supported. UPDATE: I upgrade my score on the condition that in the final copy the authors ensure that all experimental details are motivated/justified, and more importantly, all possible confounding factors are addressed, such as those brought up by R4.


Review 4

Summary and Contributions: In the context of convolutional neural networks, the paper studies the relation of training data and training-induced decision boundaries for classification objectives. For a given trained network, the authors look at the margin of a data sample as its distance to the nearest decision boundary. They propose an approximate measurement of the margin by an adversarial proxy, using DeepFool as an adversarial mechanism: What is the smallest adversarial perturbation of the sample that changes the classifier response (i.e. is on the other side of the decision boundary)? The authors use this margin approximation to hypothesize and validate on synthetic and limited real data that the margin size along a chosen subspace inversely correlates with discriminativeness of the features in this subspace.

Strengths: [S1] The paper is written well and the authors state assumptions clearly. [S2] The topic of understanding the relation of data and induced decision boundaries, including under adversaries, is a relevant topic for NIPS. [S3] The presented experiments are set up well with good attention to detail (e.g. the random basis transform $U$ to avoid potential biases from choice of basis). I liked the experiment in l167ff: flipping the DCT components between low and high frequencies. Particularly the investigation in l282-298 on the impact of changing only a chosen subset of the data seems valuable.

Weaknesses: [W1] My primary concern with the paper is that the conjecture of discriminative features causing small margins seems only tenuously substantiated. The paper and its insights are largely premised on this conjecture. The evidence towards causation is largely presented in 3.2 and I do not feel comfortable to accept it as convincing as presented. The synthetic data feels to simplistic to rule out that the presented correlations on real data have some other, unknown causes rather than supporting the conjecture. The presented insights are valuable indeed, however I wonder if causation can be evidenced on a real problem in a more acceptable way? [W2] A (stated) premise is using DeepFool as an adversary. How do I know that the observed behavior is not an artifact of the way that the margin is approximated by the adversary? Does the approximation error perhaps depend on the discriminativeness instead or is it uniform and independent of discriminativeness? Will I get the same results from a different adversary? I feel that investigating these questions would support the paper's claims. [Update after rebuttal] (1) My initial concerns around the causative factors versus the stated claims are reasonably alleviated. I suggest that the authors clarify their language around the claims in terms of cause, effect and correlation ("association") in case a casual reader of the paper has the same impression I had. (2) I do have some remaining concerns around the confounding factors in the analysis, but they do not outweigh the strengths for a conference paper. [Post-rebuttal] Thanks to the authors for the rebuttal. After considering the rebuttal: (1) My initial concerns around the causative factors versus the stated claims are reasonably alleviated. I suggest that the authors clarify their language around the claims in terms of cause, effect and correlation ("association") in case a casual reader of the paper has the same impression I had. (2) I do have some remaining concerns around the confounding factors in the analysis. In summary, I appreciate the contributions of the paper and I understand that not all confounding factors can be excluded in a conference submission. Given the revised perspective on the paper and the rebuttal, I am upgrading my initial rating.

Correctness: The methodology seems largely appropriate. There are some remaining concerns about the claims as pointed out in W1/W2 above.

Clarity: The paper is largely written well. l120-136 took a couple of readings to understand, perhaps there is some minor modification to make it clearer?

Relation to Prior Work: The paper discusses related work reasonably well.

Reproducibility: Yes

Additional Feedback: [F1] Minor clean-up: the caption in Figure 1 seems off ("top/bottom")

[Author Response · NeurIPS 2020]

We would like to thank the reviewers for their valuable feedback, which we will duly consider and integrate in our revised manuscript. In this paper, we demonstrate that "the decision boundaries of a DNN can only exist as long as the classifier is trained with some features that hold them together", i.e., DNNs have an inductive bias towards invariance. Through "a multitude of insightful experiments", "with good attention to detail", we delve into this property which sheds light on open problems like "catastrophic forgetting" and "adv. robustness". The structure of our paper (R3) is designed to (i) "rigorously confirm" the existence of this inductive bias (Sec. 3), and (ii) further investigate its consequences on the sensitivity and dynamics of adv. training (Sec. 4). Throughout the paper, we back up all our claims, first, using controlled synthetic experiments, and then, "rigorously" verifying our hypotheses on real datasets with abundant empirical evidence. We clarify the main points raised by the reviewers here below.

**Margin and features** (R4) The main claim of our paper is that DNNs *only* create decision boundaries in regions where they identify discr. features in the training data. We further shed more light on the relationship between adv. examples and features studied in [3,4]. We show that there is a big relative difference in the large margin along the invariant dirs. and the smaller margin in the discr. dirs. Nevertheless, we never claim that, within the discr. dirs, margin is at all proportional to "discriminativeness". In fact, we agree that the margin associated to different discr. features can greatly vary (Fig. 4). Overall, however, we firmly believe that the invariant dirs. will always have the largest margin.

**Causation** (R4) The main difficulty for establishing causation in our paper is the fact that the discr. features of real datasets are not known. Hence, determining their role on the geometry of a trained DNN can only be done by artificially manipulating the data. We strongly believe that the experiments in Sec. 3.2 are enough to rule out the other two main factors that might explain our results: the network and the algorithm. Specifically, in the flipping experiments, flipping the data – *ceteris paribus* – also flips the margin distribution, thus demonstrating that the margins are necessarily caused by the information present in the data. The other interventions we do on the data (e.g., low-pass experiments) confirm that in the absence of information in a certain dir. the network becomes invariant along this dir. Therefore, guided by the principles of the scientific method, and supported by strong evidence, we believe that grounds for causation are properly established.

**Choice of DCT** (R1,R3) The DCT has a long application tradition in image processing due to its good approximation of the decorrelating transform (KLT). Furthermore, in previous studies on the robustness of deep networks to different freqs., the DCT was also the basis of choice [7] because it avoids dealing with complex subspaces. A more aligned basis with respect to the discr. features would probably show a sharper transition between low and high margins. However, finding such network-agnostic bases is a challenging task without knowing the features *a priori*. The DCT is not perfectly feature-aligned, but it seems to be a good choice for comparing different architectures, especially if we compare its results to those obtained using a random orthonormal basis where differences in margin cannot be identified (c.f. Sec. N in Supp. material). We will include this explanation in the revised version of our manuscript.

**DeepFool** (R3,R4) In the adv. robustness literature DeepFool is generally regarded as one of the most efficient methods to identify minimal adv. perts. Because we measure margin, norm-constrained attacks like PGD are not suitable for our study, and more complex attacks like C&W, or using unconstrained GD in the input space, are computationally much more demanding and harder to tune while finding very similar adv. perts. to DeepFool. Hence, DeepFool is more adequate for our work. We will include this explanation in the revised version of

our manuscript. For completeness, we show a comparison of the median margin obtained on MNIST (LeNet) using DeepFool and a subspace-constrained GD attack: Even if the margins are slightly smaller for the *stronger* attack, the relative differences between regions (our quantity of interest) are the same for both attacks.

**CIFAR10 margin** (R2) Without knowing the mechanisms used by the network to select the discr. features of a dataset it is hard to give a full explanation of the low margins in the high frequencies of CIFAR10. Nevertheless, a possible reason for the different behaviour on CIFAR10 might be its low resolution. In fact, it is a common mistake that during the downsampling process no antialising filter is applied to the images before resizing, and hence some low-freq. information leaks to the high freqs. of the low-resolution images. This might explain why the network can identify some discr. features in the high-freq. spectrum of CIFAR10 (downsampling technique not specified in technical report [25]).

**Filtering other datasets** (R2) Indeed, the conclusion of the low-pass CIFAR10 experiment is generally applicable to other datasets, e.g., a LeNet trained on LP-MNIST (bandwidth of 14) has $0\%$ drop in accuracy when tested on MNIST data. We will include this result in the appendix alongside Sec. C and Sec. H (results on HP-MNIST)

**Low-pass adv. training** (R2) As seen in Fig. 7, during adv. training, the energy of the perturbations has a very small high-freq. component, and it is predominantly concentrated in the low freqs. However, it seems that the small, but non-zero components in the high freqs. of the perturbations are necessary to improve robustness, as training only using low-freq. perturbations do not yield satisfactory robustness results.

[Meta-Review · NeurIPS 2020]

This paper provides a few interesting novel insights between the discriminativeness of input features and decision boundaries. For example, the directions that an input image is nearest to the decision boundary are the directions of the discriminative features. Experiments were performed on MNIST, CIFAR-10 and ImageNet. Further findings also gave insights into how adversarial training modifies the decision boundaries to improve model robustness. Reviewers agreed that this work is particularly relevant and interesting to the adversarial example community.